# Risk Assessment for the Use of COTS Devices in Space Systems under Consideration of Radiation Effects

**Jan Budroweit** [1,*] **and Hagen Patscheider** [2]

1 Avionic Systems, Institute of Space System, German Aerospace Center (DLR), 28359 Bremen, Germany
2 Quality Management, Institute of Space System, German Aerospace Center (DLR), 28359 Bremen, Germany; hagen.patscheider@dlr.de
* Correspondence: jan.budroweit@dlr.de; Tel.: +49-421-24420-1297

**Abstract:** In this paper, a new approach is presented to assess the risk of using commercial off-the-shelf (COTS) devices in space systems under consideration of radiation effects that can dramatically affect reliability and performance. In the NewSpace era, the use of COTS has become mandatory, since typical space-qualified (class-1) electrical, electronic and electromechanical (EEE) components for space missions are no longer attractive due to their extremely high costs, long lead times and low performance. This paper sets out the usual constraints for COTS devices and proposes a guideline on how to select non-space-qualified components and when class-1 EEE devices are recommended for use.

**Keywords:** COTS; radiation effects; NewSpace; FMECA; risk assessment

## 1. Introduction

The use of commercial off-the-shelf (COTS) electrical, electronic and electromechanical (EEE) parts in space missions was often avoided in the past. The main reasons are that space systems were required to be extremely reliable and that failure was not an option because of the high mission costs and because a later repair is almost impossible. However, in the past decade, CubeSats and small satellites became more and more popular. The development of those satellites was mainly driven by universities and academia with limited budgets and restrictions on personal resources, nevertheless, with the aim of providing the same effort as for classic space missions. This in fact requires not only a different engineering approach but also the use of COTS electronics that are affordable and do not have long lead times. Both are in clear contrast to typical space-qualified class-1 devices. The consequences are higher risk acceptance and, potentially, reduced reliability, which results in reduced success rates or even the early loss of missions.

Thyrso Villela et al. showed a statistical overview of past CubeSat missions. It can be seen that, especially in the early 2000s, the success rate of such missions was very poor, as depicted in Figure 1. Especially in the first decade of the 20th century, the infant mortality of CubeSat missions was extremely high, as shown in Figure 1a, meaning that satellites failed before their first data acquisition was made. In conclusion, the success rate, illustrated in Figure 1b, was fairly low.

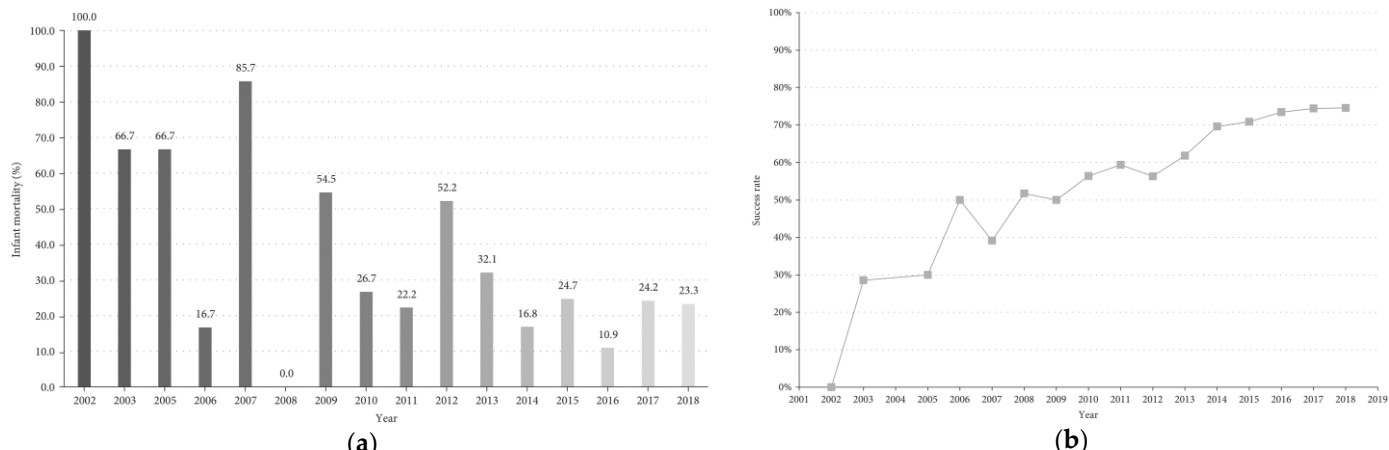

**Figure 1.** Infant mortality (**a**) and success rate (**b**) of CubeSat missions from 2000–2020. Reprinted from ref. [1].

Looking at available data, such as from the University of Saint Louis [2], it can be seen that, especially in the early 2000s, CubeSats have been developed and deployed almost exclusively from universities, as seen in Figure 2.

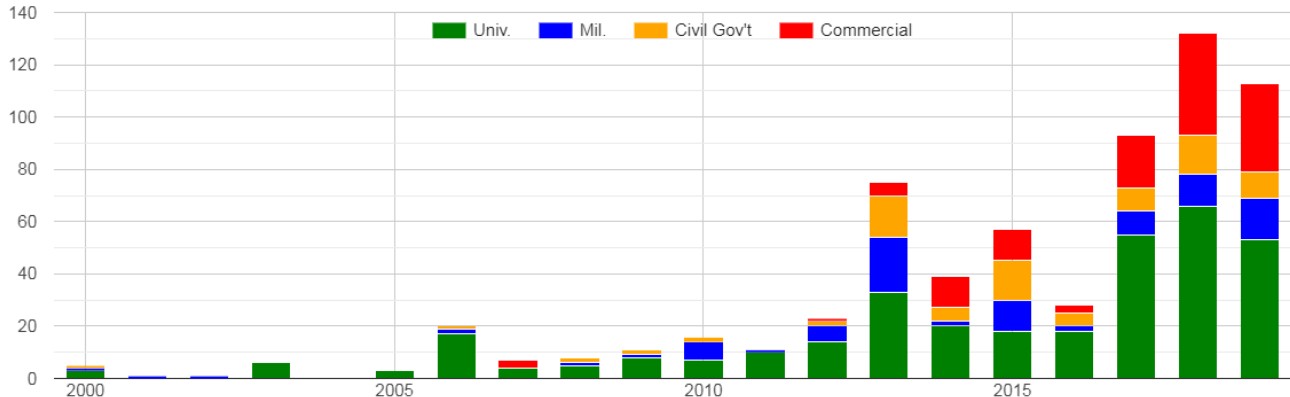

**Figure 2.** CubeSat mission types from 2000 until present, according to [2].

In fact, the low success rate is not only caused by the use of COTS EEE devices, it is more likely related to the massively reduced preparation time and the reduced mission-reliability and quality-assurance activities, which is typical for universities since the teams were mainly composed of students with limited knowledge and available time to support the mission. However, Figure 2 shows the rapidly increased numbers of CubeSat missions launched. By noting the trend towards the commercial use of CubeSats, the growing importance of nanosatellites regarding the future space market can be predicted. This trend of commercial usage of CubeSats or nanosatellites and their rapid development represents the so-called NewSpace era. The most popular example for this NewSpace trend is SpaceX with their Starlink constellation that plans to deploy over 40,000 satellites (~12,000 already approved for 2027 and up to 30,000 that are currently under approval) [3]. Obviously, such satellites cannot be built with class-1 electronic devices, just in terms of costs and the required short manufacturing time. Thus, the use of COTS parts is mandatory. The essential key for success consists of minimizing the risk of failure and improving the mission's reliability.

Since there are many concerns about COTS devices, especially for use in the harsh environments of space, system designers will have to carefully select EEE components, and risk assessment is therefore crucial. Currently, there are neither dedicated standards nor guidelines for the use of COTS in space missions. Engineers need to follow their

own judgment. This paper proposes an approach for risk assessment for the use of COTS devices in space systems and a potential guideline for their selection.

### 1.1. Concerns about Using COTS in Space

The use of COTS devices for critical applications has a long history. In 1994, William Perry's directive as U.S. Secretary of Defense officially initiated the use of COTS in military applications, which usually follows similar requirements as that of space applications. For many space applications, COTS are just alternative options if costs and performance are key drivers. Especially in terms of costs, space-grade devices are often 1000 times more expensive compared to COTS alternatives. The main reason is that the whole screening and evaluation flow for class-1 EEE devices requires tremendous effort, as can be seen in the evaluation flow from the European Cooperation for Space Standardization (ECSS) of the ECSS-Q-ST-60-13 presented in Figure 3.

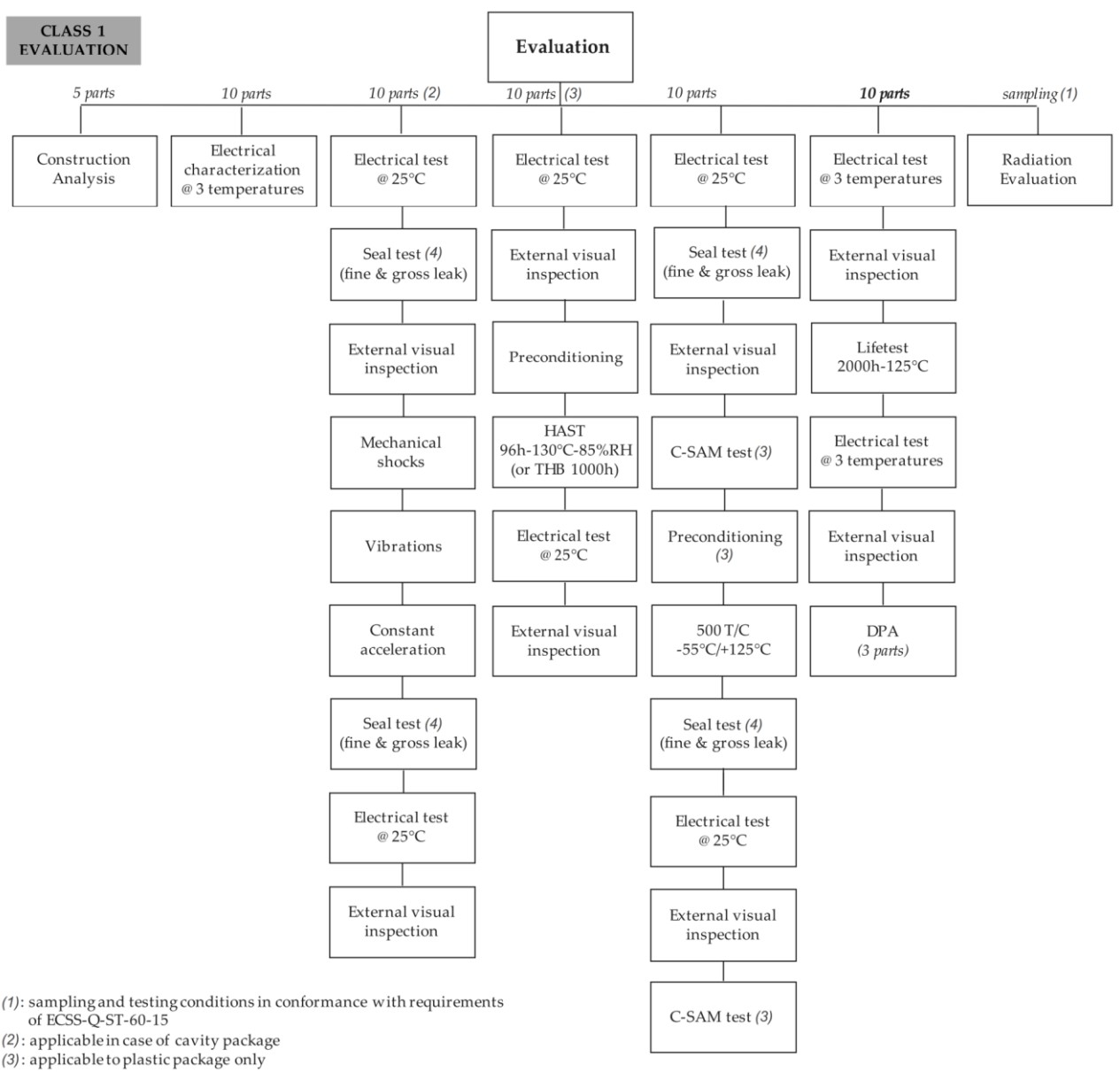

**Figure 3.** Evaluation tests flow chart for class-1 EEE components. Reprinted from ref. [4].

However, some of those extreme test sequences are potentially over-rated for space components. For instance, the thermal stresses that are applied for the screening flow ($-55\,°C$ to $+125\,°C$) are in most cases way above the ratings that are achieved inside of a spacecraft.

Another environmental condition that seems to be less critical for EEE parts is the mechanical shock and vibration loads, which usually appear for a very short duration during launch and separation from the rocket.

One of the main issues with COTS is that the ratings and qualification levels can vary from manufacturer to manufacturer. Product traceability is often not guaranteed, which results in unknown statistics (e.g., infant mortality, updates of the product lines, or changes in the fabrication process) of devices that are available off-the-shelf. With higher qualification levels, such as automotive or defense/military grades, information about the fabrication site, the date code and lot number are often provided, also through distributors. Moreover, automotive parts qualification, according to AEC-Q100, follows a similar screening process presented in Figure 3. The only missing test branch is the radiation effects evaluation, which in fact is the most critical environmental condition in space and cannot be neglected. For this reason, this paper primarily aims at radiation effects—including total ionizing dose (TID) and single-event effects (SEE)—and their risk assessment for space system design when the use of COTS shall be considered.

### 1.2. Failure Mode, Effects and Criticality Analysis

Spacecraft are exposed to harsh environmental conditions. Extreme temperature ranges, the vacuum of space and high-energy radiation in combination with the impossibility of maintenance and replacement measures make the space environment particularly challenging. Spacecraft therefore particularly depend on reliable components.

### FMECA

It is quite evident, that risk analyses are of great importance in space projects. A failure mode, effects and criticality analysis (FMECA) is a tool designed to systematically identify potential failures in products and processes and to assess their effects [5]. The FMECA builds the base for the definition of risk mitigation strategies. In space projects, it is used in particular to define the failure tolerance design, to give special test recommendations and to set operational constraints [5]. The FDIR (fault detection, isolation and recovery) concept is an important cornerstone of the failure tolerance design within a spacecraft. It is one of the key functionalities of the on-board software and relies mostly on a failure hierarchy provided by the FMECA that specifies on which level failures are to be fixed [6,7].

Consequently, the FMECA is an often-used tool in space projects. The ECSS has introduced a dedicated standard referring to FMECA principles and requirements. FMECA is executed as a bottom-up analysis, wherein the effects of the identified failure modes are followed up to the boundaries of the product or process under investigation [5]. The depth of the analysis is limited to the point wherein meaningful recovery strategies can be defined [8]. According to the ECSS, the FMECA is performed as follows:

1. Description of the product or process under investigation
2. Identification of all potential failure modes for each item
3. Taking the assumption that each failure is the only failure in the product; combinations of failures are therefore not considered
4. Evaluation of the failure modes as a worst-case scenario and determination of the criticality number
5. Identification of failure detection modes
6. Identification of existing preventive measurements
7. Providing actions to correct the failures for identified critical items
8. Documentation of the analysis
9. Recording all identified critical items in the critical items list.

The criticality ranking in step 4 is one of the main tasks when executing a FMECA. The criticality number for a specific failure mode derives from the severity of the failure effect and the probability of the failure mode occurrence. In case of the process FMECA, the criticality number (CN) is defined as the product of the assigned failure mode severity (severity number SN), the probability of occurrence (probability number PN) and the probability of detection (detection number DN) as follows:

$$CN = SN \times PN \times DN \tag{1}$$

The severity number of a failure mode refers to different severity categories associated with different severity levels. The presence of redundant hardware does not affect the severity assessment of a failure mode. If the failure mode can have more than one failure effect, the highest SN will be considered. A critical step when assessing the severity number is the determination of the failure effects resulting from the defined failure. In space projects, the project team will agree on a definition of failure effects. Table 1 shows an example based on the ECSS.

**Table 1.** Severity numbers (SN) applied at the different severity categories with associated failure effects.

| Severity Level | Severity Number (SN) | Severity Category | Failure Effect |
| --- | --- | --- | --- |
| 1 | 4 | Catastrophic | Propagation of failure to other systems, assemblies or equipment |
| 2 | 3 | Critical | Loss of functionality |
| 3 | 2 | Major | Degradation of functionality |
| 4 | 1 | Negligible | Minor or no effect |

The second step when calculating the CN of a failure mode is the estimation of the probability number. The probability of occurrence for dedicated failure modes is known for certain components. There are many available tools for the reliability prediction of components. The military standard MIL-HDBK-217F is a still widely used guideline for predicting reliability in space programs. Reliable information can also be found in databases like NPRD-2016, FMD-2016, SPIDR or OREDA [8]. The qualitative approach based on "best engineering judgment" is used when no data are available. The project team must agree on probability levels. Table 2 shows an example based on [7].

**Table 2.** Probability levels, limits and numbers.

| PN Level | PN Limits | PN |
| --- | --- | --- |
| Very likely | $P > 1 \times 10^{-1}$ | 4 |
| Likely | $1 \times 10^{-3} < P \leq \times 10^{-1}$ | 3 |
| Unlikely | $1 \times 10^{-5} < P \leq \times 10^{-3}$ | 2 |
| Very unlikely | $P \leq 1 \times 10^{-5}$ | 1 |

The estimation of the detection number represents the third step when calculating the criticality number. It affects the criticality of a failure by considering potential detection and recovery processes [4]. Table 3 shows an example of detection numbers and their definition based on [7]:

**Table 3.** Detection numbers and definition.

| DN | Definition |
|---|---|
| 4 | Very unlikely |
| 3 | Unlikely |
| 2 | Likely |
| 1 | Very likely |

This estimation can be done by means of a qualitative approach based on experience. The quantitative approach may be difficult to apply in space projects, where small numbers of products or prototypes are developed. However, specifically for radiation effects, certain strategies for detection and recovery can be implemented, such as memory scrubbing or current sensing nodes, that improve criticality due to the probability of detection.

The DN adds a third dimension to the CN that can be calculated according to Equation (1), once all three numbers are known. The CN takes on a value between 1 and 64. An example of a criticality classification can be found in [4] with a criticality limit or threshold to be defined. The CN's are shown as a three-dimensional criticality matrix, distinguished by the four levels of the detection number (see Figure 4).

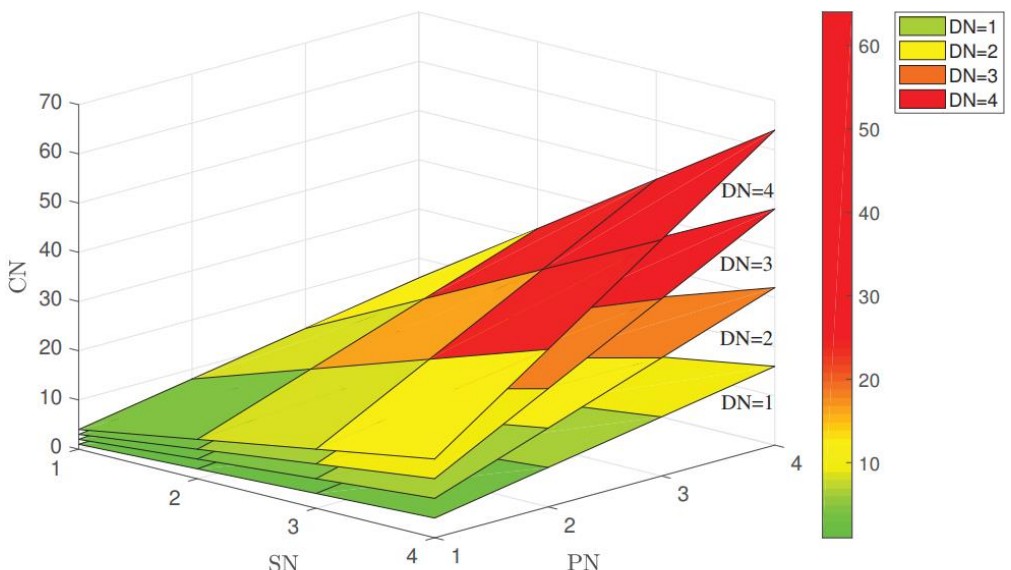

**Figure 4.** Criticality number (CN) matrix with applied limits.

To identify critical items, the project team has to agree on a criticality ranking. The classification depends on the need of the project and the acceptable risk of the mission.

The FMECA is a risk-analysis method independently applicable to various failure modes. However, its direct application in single-event analysis seems very infeasible. This is especially problematic because of the growing importance of SEE analyses for spacecraft. In many cases, it is neither affordable nor possible to develop SEE-immune hardware [6]. To respond to this situation, Gates et al. [6] developed an approach tailored for the analysis of SEEs and their propagation—the so-called single-event effect criticality analysis (SEECA). The SEECA is a system engineering approach, which combines the analytical approach of dependability analysis (like FMECA) with the special knowledge needed to cope with SEE. The main steps in performing an SEECA are [7]:

- Function analysis
- A functional analysis of the system provides the foundation for studying the impact of SEEs
- Single-event effect perspectives

- Investigating different design options to mitigate SEEs and meet the required performance at the same time
- Functional criticality
- Functions are categorized into "criticality classes", or categories of differing severity of SEE occurrence
- Functional and device SEE requirements
- Definition of SEE-requirement strictness: the more critical a function is the stricter the SEE requirement should be.

Similar to the FMECA, one of the SEECA's cornerstones can be seen in the criticality ranking of the functions (see functional criticality, bullet #3).

The here-presented risk assessment approach for the use of COTS devices in space systems, considering radiation effects, follows the main rules of the FMECA and SEECA and highlights guidance for the selection of COTS devices in space applications.

## 2. Risk Assessment Approach

Radiation hardness assurance (RHA) for using COTS devices is not a new topic. It has already been covered over several years and certain publications can be found, e.g., [9–12] G. L. Hash et al. [9], already proposed in 1997 different categories of integrated circuits, from radiation-hardened to radiation-tolerant, to COTS devices. The main differences from a manufacturing point-of-view have been introduced and the main concerns regarding COTS that need to be considered by system designers have been outlined. Radiation hardness levels were proposed for all categories based on experimental results for different technologies. Mission requirements, such as lifetime or orbital specification, that have a big impact on the radiation environment were not considered in [9]. Campola published an RHA process that allows a determination of the system needs using the given environment and mission requirements [10]. Based on these assumptions, Campola proposed a quantified risk for using EEE devices and what data information would be required, as depicted in Figure 5.

**Figure 5.** Radiation data required for quantifying risk in a represented mission and different mission durations. Reprinted from ref. [10].

Comparing the table presented in Figure 5 with the FMECA-based criticality ranking as illustrated in Figure 4, the similarity of both approaches can be noticed. However, the RHA process used by Campola is primarily related to engineering judgment and does not

cover an analytical way of determining risk assessment with a proper value. The novelty of the herein-proposed risk-assessment approach is that it allows a value-based criticality ranking instead of using an engineering-judgment-only approach.

A Bayesian method for bounding SEE-related risk and TID RHA has been published in [11,12]. The Bayesian RHA approach has been developed by Ladbury et al. to improve risk mitigation using a broad set of data. The Bayesian method describes how the probability of occurrence changes when new data are considered. It allows designers to determine the risk when using COTS devices and to especially consider especially lot-to-lot variations and the probability of failure occurrence. Thus, it allows designers to decide whether it is mandatory to upscreen/test devices again or to use the EEE devices based on the given set of data. Comparing this approach with the one being presented in this paper, the method of Ladbury et al. focuses more on single devices and does not consider a system point-of-view. It does not include potential failure propagation and risk assessment. This circumstance can be seen as a novelty of the herein-proposed risk-assessment approach.

In the following, this new FMECA-based RHA approach will be explained and is separated into different stages/steps:

Step 1: System level breakdown structure into functional block design
Step 2: FMECA-based severity analysis performed on functional blocks
Step 3: Technology assessment and rating on functional blocks
Step 4: Evaluation of the FMECA-based criticality of selected devices.

### 2.1. System Level Breakdown

In order to reduce the complexity of the proposed risk-assessment approach, the desired system is firstly seen as a black box without any specific knowledge about its intended function. In the second step, a functional block design is created (*step 1*). The black box design is depicted in Figure 6.

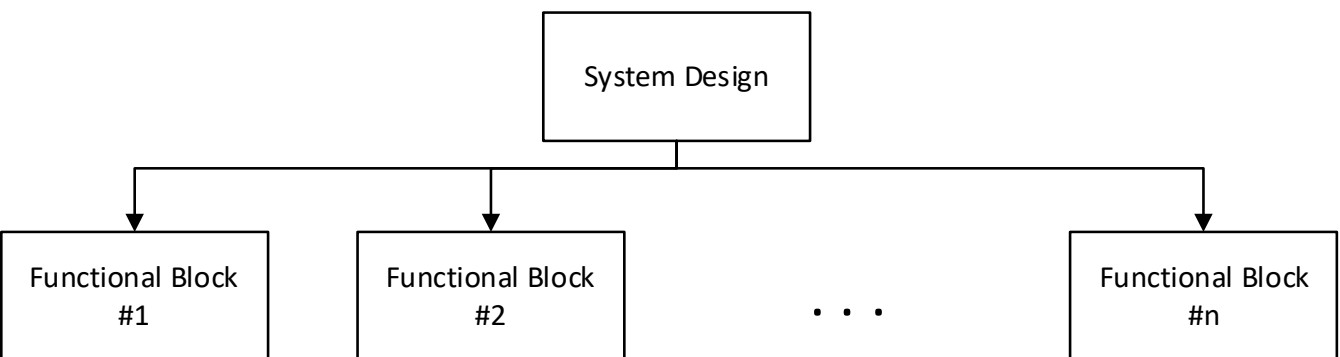

**Figure 6.** Breakdown structure of a functional block design.

The block design shall cover any functionality of the black box design and separate those into dedicated functional blocks. The reason for this is to identify the criticality of each block and to perform a FMECA severity analysis to assess the risk for the overall mission/spacecraft (*step 2*) and later select corresponding device(s) and quality grade(s). Usually, a FMECA can be structured down to individual devices, but this requires that certain components have already been selected.

A technology assessment and rating is performed in order to compare different criteria that need to be taken into account for the later selection process of potential electronic devices (*step 3*) once the FMECA has been performed. The technology assessment is introduced in Section 2.3.

If potential technologies and devices have been reviewed, rated and selected, the final evaluation for criticality analysis and acceptability of these devices is made (*step 4*). The specific evaluation method and device selection process is presented in Section 2.4.

### 2.2. FMECA-Based Severity Analyis for Radiation Effects

A dedicated FMECA-based severity analysis will be performed on each functional block to evaluate the severity of potential failures. In this proposed risk assessment, the focus is based on radiation effects that cause different failure modes and resulting effects, and then, the severity is determined.

### 2.3. Technology Assessment

As there might be different technologies and devices for the block(s) available to fulfill the functional requirements, a technology assessment is useful prior to the selection of a specific device. For this process, different criteria can be reviewed to rate the potential technology and device candidates:

1. Level The level displays the different available qualification levels of intended devices or technologies.
2. Review The review rates the available data that is provided by the manufacturer, including product traceability, quality assurance documentation or product change notifications.
3. Complexity The complexity of the intended technologies or devices can differ greatly, which later has a direct impact on handling and implementation, e.g., software code development and compilation. For instance, the complexity of an FPGA is essentially higher than that of a bipolar transistor. In general, a lower complexity has a better rating.
4. Performance Besides the costs, performance is mainly why COTS parts are preferred. However, performance can differ strongly between technologies and is thus a relevant criterion for technology assessment.
5. Costs As described in the aforementioned performance criterion, costs are one of the essential drivers for development of space systems.
6. Data Especially when space-qualified class-1 EEE components are not available or not desired, available data or information for environmental stress response, in particular for radiation, of the technology or device is required.

The criteria shall be used and dedicated ratings be applied, e.g., from poor (–) to excellent (++). With this rating, design and test engineers can pick up the most suitable devices and perform the following criticality evaluation, as discussed in Section 2.4.

### 2.4. Criticality Evaluation and Device Selection Method

In this section, the criticality evaluation flow for intended devices' functional blocks is presented. The workflow is depicted in Figure 7 and firstly, follows the evaluation of the severity number (SN) gained by the block-related FMECA. As introduced in Section 1, the highest (SN) of four (4) is associated with catastrophic failures wherein propagation to other external systems, assemblies or equipment is expected. Such failure propagation could lead to a complete loss of a mission.

Thus, if one expects failure results with a SN of four (4), the use of COTS is not recommended and a radiation-hardened (RadHard) class-1 device should be used. If there is no class-1 solution available, a COTS alternative needs to be considered.

For a SN less than four and greater than or equal to three ($4 < SN \geq 3$), COTS are generally worth considering. However, since there are certain concerns with the use of COTS as discussed in Section 1, a detailed manufacturing review needs to be done. The manufacturing review process is divided into *mandatory* and *desirable* branches. In case of a SN of three, both reviews have to be passed before a further investigation is acceptable. During the mandatory review, specific requirements shall be verified. This includes that the manufactured devices follow certain quality assurances (e.g., ISO9001), that a process monitoring control is given and that a notification service about product changes will be provided to the customers. This information is important to verify whether gained or referenced data is still valid or if the product is becoming obsolete. The desirable review shall guarantee that the product that is available has a high qualification grade,

which implies that it underwent a deeper screening process. As a baseline, automotive-grade (AEC-Q100) components are mandatory to use. Higher screening levels, such as for defense/military-grade parts, are desirable.

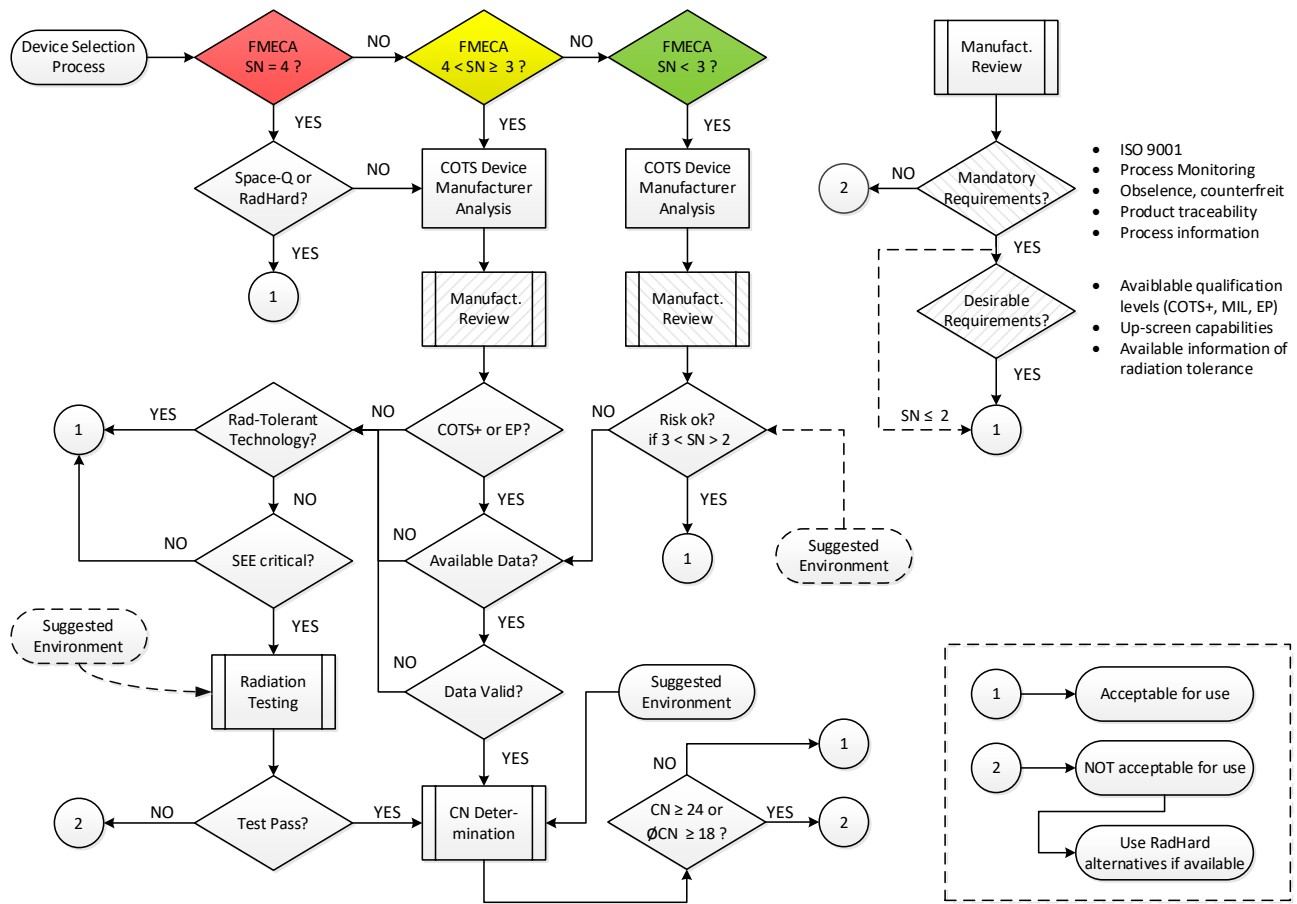

**Figure 7.** EEE component evaluation and guidance flow chart.

Once the manufacturing review is passed, it is necessary to analyze whether radiation hardness data is available for the selected device. In some cases, institutions such as NASA or ESA provide and publish a wide portfolio of already radiation-tested COTS devices.

Those data are mostly publicly accessible and can be used as a reference. Nevertheless, it shall be verified that the data is still valid by checking the tested lot numbers and product change notifications of the manufacturer. If no information about radiation hardness is available, radiation testing becomes necessary. Depending on the device technology, the effort of radiation testing can be limited, e.g., if the SEE susceptibility is not given and only total dose testing needs to be applied. For some technologies such as for wide-band gap semiconductors, e.g., gallium–arsenide (GaAs) or gallium–nitride (GaN), total ionizing dose is not a concern. However, some technologies have been shown to be susceptible to SEEs; thus, a specific risk assessment needs to evaluate if SEEs might be critical or not. When it comes to radiation testing, there is no need to follow the complete evaluation flow according to standards such as ESCC-25100 [13] or ESCC-22900 [14]. At this stage, the suggested environment, e.g., low Earth orbit (LEO) can be used as an input to define the test requirement that potentially reduce the time for testing and resulting costs.

For severity numbers less than three (3), only a mandatory manufacturer review is required. It is still recommended to choose at least automotive-grade devices if available. Those devices are extensively screened, compared to industrial-grade devices, and infant mortality is deeply considered. In terms of radiation effects and hardness assurance, those devices do not have to undergo the previously mentioned evaluation. However,

depending on the mission requirements, further risk assessment can be made, considering the suggested environment before their acceptance for use is defined.

Finally, if data from references can be used, or if own test data has been generated, a criticality assessment can be performed in order to evaluate acceptable use for the intended device. A criticality number is calculated according to Equation (1) in Section 1.1. For the CN determination, the probability number (PN) can be derived from in-orbit error-rate predictions that can be calculated with the gained cross-sections of the test results and with the expected environmental fluxes of particles. Tools such as OMERE [15] can be used to predict those in-orbit error rates. Since failures can be detected through system-level mitigation techniques or failure isolation mechanisms, the criticality number can be improved. The impact of error detection is expressed by the detection number (DN) and can be considered if the effectiveness of the mitigation mechanisms is known or even using best-practice experience and engineering judgment.

The CN determination is performed on each expected failure for the functional block as defined by the FMECA evaluation. The threshold for the criticality number can be selected according to the mission and quality assurance requirements. For instance, as expressed in Figure 7 (CN determination), if the criticality number of each failure does not exceed 24, and the average CN for the functional block is less than 18, the device is acceptable for use.

It has to be mentioned that the proposed RHA approach and guidance scheme presented in Figure 7 mainly focuses on TID and SEEs. Displacement damages (DD), which can have a big impact, especially to optoelectronics, have not been covered in this paper. However, the FMECA-based risk-assessment approach can also be applied to those technologies by considering DD as an additional source of failure.

### 3. Implementation and Discussion

In order to demonstrate the proposed risk-assessment approach and selection guidance presented in Section 2, a simplified data-handling system is chosen as an example and will be described in this section.

### 3.1. Functional Block Design

Starting with the functional block separation, the data-handling system consists of the blocks depicted in Figure 8.

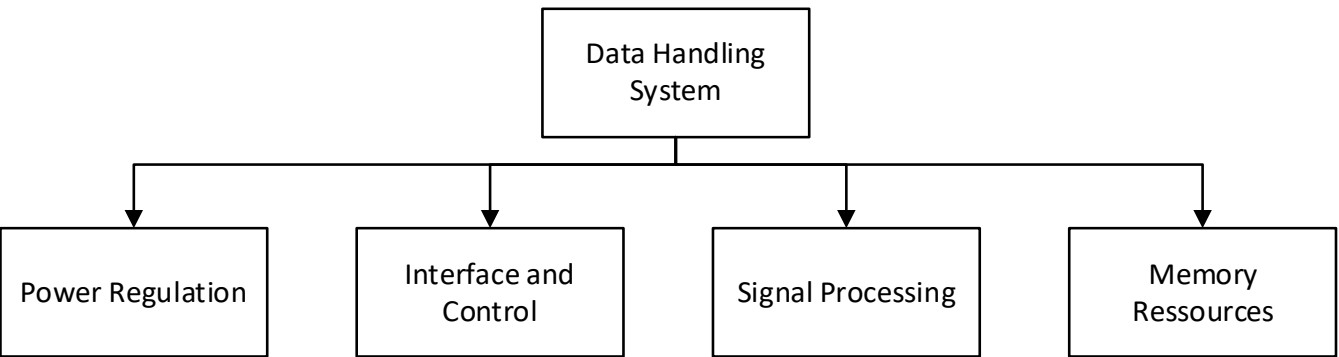

**Figure 8.** Simplified functional block design of a data-handling system.

Further functional blocks can be implemented into this breakdown structure or the given block can be described/separated in more detail.

- Power regulation This block represents all required power electronics, such as isolated DCDC converters for the external power supply, as well as a system-internal buck converter or low-dropout regulators.

- Interface and control The interface and control block represents the external and internal electrical interfaces of the data handling system, such as RS422, low-voltage differential signals (LVDS) or the ethernet.
- Signal processing The signal-processing block consists of electronics that are responsible for the data and signal processing of the system (e.g., capturing and execution of commands or generation of telemetry data). Digital signal processors (DSP) or field-programmable gate arrays (FPGA) are potential technologies that can be used for digital data processing.
- Memory Resource This block could consist of either static memory devices to store nonintermediate data or dynamic memory devices to provide computing resources, e.g., by synchronous dynamic random-access memory (SDRAM) technology to the signal-processing device.

### 3.2. FMECA-Based Severity Analysis

Based on the functional breakdown structure, the FMECA is applied to each of these blocks to identify potential failure modes, their impact and their severity. Two examples are presented for the data-handling system: (1) the interface and control block and (2) the signal-processing block. Again, since this paper primarily aims to cover radiation effects, only failures caused by radiation effects are considered. However, the FMECA severity evaluation could also cover other expected defects and failures.

#### 3.2.1. Interface and Control (CTRL)

The FMECA evaluation for the CTRL block is presented in Table 4. Different types of expected radiation effects and their resulting failures are considered, including SEEs and the TID. The assigned severity numbers are shown and their corresponding failure effect can be derived.

**Table 4.** FMECA result for the interface and control block.

| ID | Failure Mode | Failure Cause | Failure Effect | SN |
|----|----|----|----|----|
| CTRL-1 | HW | High current state (SEL) | catastrophic failure affecting external systems | 4 |
| CTRL-2 | HW | Long-term degradation (TID) | catastrophic failure affecting external systems | 4 |
| CTRL-3 | HW | Critical (voltage) transients (SET) | catastrophic failure affecting external systems | 4 |
| CTRL-4 | HW | Long-term degradation (TID) | permanent loss of system functionality | 3 |
| CTRL-5 | HW | Critical (voltage) transients (SET) | permanent loss of system functionality | 3 |
| CTRL-6 | HW | Long-term degradation (TID) | permanent loss of system functionality | 3 |
| CTRL-7 | HW | Non-critical (voltage) transients (SET) | corrupted data transmission/ interpretation | 2 |

According to the presented device selection flow in Figure 7, a SN of four (4) leads to the recommendation of a RadHard solution (if available), because failure propagation to external systems is expected (e.g., the satellite bus) and could lead to a complete loss of the mission. In this case, all interface devices, e.g., RS422 driver or receiver, are preferred for a class-1 EEE part category.

### 3.2.2. Signal Processing (SP)

The FMECA severity analysis for the signal-processing block is presented in Table 5. Since signal processing is not directly affecting external systems (e.g., by electrical connections), no severity number greater than three (3) is determined. Thus, COTS devices are, in general, acceptable. However, potential devices that can replace this functional block need to follow the procedure presented in Figure 7, including the manufacturing review and the verification of available radiation test data or undergoing a new radiation evaluation.

**Table 5.** FMECA result for the signal-processing block.

| ID | Failure Mode | Failure Cause | Failure Effect | SN |
|---|---|---|---|---|
| SP-1 | HW | High current state (SEL) | permanent loss of system functionality | 3 |
| SP-2 | HW | Long-term degradation (TID) | permanent loss of system functionality | 3 |
| SP-3 | HW | Non recoverable (stuck) state (SHE) | permanent loss of system functionality | 3 |
| SP-4 | HW | Recoverable loss of function (SEFI) | corrupted data transmission/ interpretation | 2 |
| SP-5 | SW | Crash of operating system (SEU/MBU/SEFI) | corrupted data transmission/ interpretation | 2 |
| SP-6 | SW | Crash of software/applications (SEU/MBU/SEFI) | temporary loss of system-parts' functionality | 1 |

To continue the proposed example, a technology and device assessment for the signal processing is performed and discussed in Section 3.3.

### 3.3. Technology Assessment

The technology assessment for a signal-processing block is shown in Table 6. For this assessment and survey, fictive data have been used to avoid advertising. Different technologies for the signal-processing block are considered, including DSPs, application-specific integrated circuits (ASIC), FPGAs or system on chips (SoC). The first rating factor (level) shows the maximum available qualification grade. The factor "review" shows information available from the manufacturer, which is more applicable once a certain technology has been chosen (thus not applicable (n.a.) for technology assessment). Other factors like complexity, performance, costs and data are compared and evaluated for the investigated technologies. Available data are more precise for dedicated devices out of the selected technology family but can be also covered for the technology assessment and rated (similar to the review assessment). The technology assessment for the signal-processing block shows that the SoC is the most promising technology, since the balance between cost, performance and complexity is good and sufficient data would be available for reference.

**Table 6.** Technology assessment for the signal-processing block.

| Device | Technology | Level | Review | Complexity | Performance | Costs | Data |
|---|---|---|---|---|---|---|---|
| DSP | CMOS | Industrial | n.a. | ++ | - | ++ | -+ |
| ASIC | BiCMOS | Space | n.a. | - | ++ | − | n.a. |
| FPGA | CMOS | Automotive | n.a. | -+ | + | + | + |
| SoC | CMOS | Military | n.a. | - | ++ | + | ++ |

In the next step, dedicated SoC devices are compared. As an example, Table 7 provides a set of SoC devices currently available on the market. It should be noted that the assessment factor "review" is applicable now that a technology has been selected.

**Table 7.** Device assessment for the signal-processing block.

| Device | Technology | Level | Review | Complexity | Performance | Costs | Data |
|--------|-----------|-------|--------|-----------|-------------|-------|------|
| A | 28 nm CMOS | Military | + | -+ | + | ++ | ++ |
| B | 130 nm CMOS | Military | -+ | -+ | -+ | ++ | -+ |
| C | 28 nm CMOS | Automotive | - | - | + | + | - |

Based on the rating performed on three candidates, device A is the most promising candidate since military-grade quality is available, resulting in a higher screening and device reliability. Considering the availability of radiation test data (++) that can be accessed, device A shows the best results. At this stage, it is assumed that the radiation test data of device A is valid (as required by the selection flow in Figure 7) and can be used for the last step, the determination of criticality and for the final decision if this device is acceptable for use.

### 3.4. Criticality Evaluation

In this section, the criticality of the intended device A for the signal-processing block is determined according to the selection flow depicted in Figure 7. The device has passed the manufacturing review (both mandatory and desirable) and is available in military-grade quality that implies a set of evaluation test standards as well as given fabrication and process traceability. For the device, a broad set of valid radiation test data are available. Table 8 summarizes them for SEEs with predicted in-orbit event rates for a 2 year LEO mission under nominal and worst-case conditions. As discussed in Section 2.3, tools such as OMERE can be used to predict in-orbit rates with available cross-section and linear energy transfer (LET) threshold data that is used for the determination of the probability number (PN).

**Table 8.** Predicted in-orbit rates for different types of SEEs based on the radiation test data and their cross-sections.

| SEE Type | LET Threshold $(MeV \cdot cm^2/mg)$ | Cross-Section $(cm^2/bit;dev)$ | Event Rate/Day (Nominal) | Event Rate/Day (Worst) |
|----------|-------------------------------------|-------------------------------|--------------------------|------------------------|
| SEL | $2.1 \times 10^2$ | $3.1 \times 10^{-4}$ | $5.1 \times 10^{-5}$ | $1.4 \times 10^{-3}$ |
| SEU/MBU | $4.2 \times 10^1$ | $2.2 \times 10^{-9}$ | $1.6 \times 10^{-8}$ | $3.2 \times 10^{-7}$ |
| SHE | $3.7 \times 10^3$ | $1.1 \times 10^{-10}$ | $7.2 \times 10^{-14}$ | $4.2 \times 10^{-12}$ |
| SEFI | $2.5 \times 10^1$ | $9.5 \times 10^{-3}$ | $3.2 \times 10^{-2}$ | $6.2 \times 10^{-2}$ |

Similar tools can be used to estimate the total dose over the mission duration. Typical values for a 2 year LEO mission (depending on the altitude and inclination) are 10 krad($SiO_2$). This value also strongly depends on the surrounded shielding and is expected to be lower inside of the spacecraft (in a range of five to seven krad($SiO_2$)). Test results on the intended device A showed no significant degradation or malfunctions up to total doses of 100 krad($SiO_2$).

To finally determine the criticality, based on the available radiation test data and the derived predicted in-orbit rates, Equation (1) is applied to calculate the criticality number of each expected failure according to Table 5 (for severity numbers). For the PN, the worst-case event rates are used. For instance, SELs were predicted to occur every 715 days $((1.4 \times 10^{-3})^{-1})$, which for a 2 year LEO mission is extremely low, once per mission duration (PN = 1). The detection number is selected according to potential implemented

mitigation strategies. For instance, high current events as a result of an SEL can be detected through the measurement of the current conditions on dedicated power lines.

Other techniques can also be applied for dedicated failure causes, but not all of those effects and failures might be detectable. According to Table 9, the maximum CN that has been determined is 18 for SEFIs on the hardware and software side. The average CN for the whole device is calculated to be 8.57. Related to the thresholds as defined in Figure 7, the device would be acceptable for use. However, depending on the mission and quality assurance requirements, the threshold can be adapted if the risk acceptance is lower or higher.

**Table 9.** Criticality determination for the signal-processing device A.

| ID | Failure Cause | SN | PN | DN | CN |
|----|---------------|----|----|----|----|
| SP-1 | High current state (SEL) | 3 | 1 | 2 | **6** |
| SP-2 | Long-term degradation (TID) | 3 | 1 | 2 | **6** |
| SP-3 | Nonrecoverable (stuck) state (SHE) | 3 | 1 | 2 | **6** |
| SP-4 | Recoverable loss of function (SEFI) | 2 | 3 | 3 | **18** |
| SP-5 | Crash of operating system (SEU/MBU/SEFI) | 2 | 3 | 3 | **18** |
| SP-6 | Crash of software/applications (SEU/MBU/SEFI) | 1 | 3 | 2 | **6** |
| | | | **Average CN:** | | **8.57** |

## 4. Conclusions

In this paper, a new risk-assessment approach has been presented for the use of COTS EEE parts in space systems, primarily under consideration of radiation effects. The proposed methodology uses the FMECA technique and presents selection guidance for the use of EEE devices. Therefore, the system is firstly subdivided into functional blocks, for which an independent severity classification is made to identify their potential risk to the mission and the failure propagation to other systems (e.g., the satellite bus). If the severity determination allows the potential use of COTS devices, a follow-on procedure is presented. This includes a technology and devices assessment suitable for the functional blocks and a criticality analysis based on available reference data or on own investigated radiation test data. Finally, the criticality determination shall help designers and engineers to decide if the intended and analyzed COTS part is acceptable for use, if alternative solutions need to be evaluated or if they should use class-1 RadHard devices

Risk assessment is a critical point of this study and needs to be tailored to the specific (mission) requirements. However, the presented approach allows system designers to use COTS devices based on analytical data instead of applying only engineering judgment and best-practice experience as was usually done. Especially with the growing importance of the CubeSat market and the NewSpace era, the guidance and selection approach can be helpful to minimize risk and to avoid extraordinary mission costs.

**Author Contributions:** Conceptualization, J.B.; fundamentals, H.P.; data curation, J.B.; validation, J.B. and H.P.; analysis, J.B.; writing—original draft, J.B. and H.P; writing—review & editing, J.B. All authors have read and agreed to the published version of the manuscript.

**Funding:** This research received no external funding.

**Conflicts of Interest:** The authors declare that there is no conflict of interest regarding the publication of this paper.

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
