# Peer review of "Risk Assessment for the Use of COTS Devices in Space Systems under Consideration of Radiation Effects"

_electronics, doi:10.3390/electronics10091008_

Round 1

Reviewer 1 Report

I wanted to thank the author for how well presented and clear everything is, it is not always possible to find papers in such good conditions.

On the other hand, and although the structure seems good to me, I think that points 2 and 3 could be part of the introduction. It is not until point 4 that this new approach for the selection of COTS components in space is presented.

I wanted also to point out that the effects of displacement damage, which are the most important when talking about optoelectronic components, are completely left out. I understand that the work leaves out these technologies, but it could at least mention these effects and these technologies and say that this approach would also be valid for evaluating this type of devices.

In Figure 5: "#1" appears twice instead of "#1" and then "#2".

Author Response

Dear reviewer,

thanks for taking time and reviewing our submitted paper. Your effort and comments are highly appreciated. We revised the manuscript according your comment and suggestions and responded to them by the attached document.

Thanks again and kindly regards

Reviewer 2 Report

This paper presents a new approach to assess the risk for use of COTS devices. Some minor concerns are provided below. (1) What is the novelty of this paper. The authors should further justify the novel contributions of this work. (2) The authors could make a comparison between this approach and those reported ones. (3) The writing should be improved. For example, figs. 1 and 2 are not clear.

Author Response

Dear reviewer,

thanks for taking time and reviewing our submitted paper. Your effort and comments are highly appreciated. We revised the manuscript according your comment and suggestions and responded to them by the attached document.

Thanks again and kindly regards

This manuscript is a resubmission of an earlier submission. The following is a list of the peer review reports and author responses from that submission.

Round 1

Reviewer 1 Report

The manuscript deals with one of the hot topics of new space (economy), that is finding new methods to assess the possible use of COTS in an objective way.

The paper proposes a way to deal with this evaluation process, and could give an interesting contribution to the discussion on this topic.

For this reason it would be desirable to explain your proposed method in the clearest possible way. To this aim I propose below some possible improvements to an already well written paper:

Row 48: applied->under approval?

Row 50: “and therefore are”

Row 100: even if known in the field, please define the ECSS acronym 

Row 147 table 3: it seems that there is a mistake in the definition of either DN 4 or DN 1

Row 181: I would suggest “ radiation effects, “ (add comma) and “highlights” 

Row 192: there is some word missing after “)” I guess

Rows 206, 211, 241: it would be better to match the numbered list of steps at row 186, by adding a subsection 4.1 about “system level breakdown” on row 191 and shifting the others. This would make the method more clear to the reader.

Row 238: better to stress that “excellent” is related to quality and not to quantity. E.g. an “excellent” for the cost criterion means “cheap” and for the “complexity” means “simple” (at least this is what I understand)

Row 344 table 4: ctrl 4 and 6 seems to be duplicated, could be that 6 should have SN=2?

Rows 364-365: it could be better to separate these two steps also in the general description of the technology assessment, since it would happen frequently to have to perform the two steps separately.

Rows 379-381: I would move this consideration to the next paragraph while discussing the criticality evaluation 

Row 383: I would here follow the path along the flowchart, for better clarity for the reader since you are explaining an example of application of your method. In the text you go directly to the final step about the CN evaluation instead.

Row 400: power->per, I guess. 

Author Response

Dear Reviewer,

first, we would like to thank you for reading and reviewing our paper. Your comments and suggestions are highly appreciated. The detailed replies can be found in the attached word file.

Reviewer 2 Report

Title : Risk Assessment for the Use of COTS Devices in Space Systems under Consideration of Radiation Effects
Authors : Jan Budroweit and Hagen Patscheider
-----------------------
In this paper, the authors present a new approach to assess the risk for the use of commercial-off the-shelf (COTS) devices in space system under consideration of radiation effects.  

Strengths :
. The subject addressed and the method, clearly presented.

Weaknesses :
. The bibliography.
. The data, to illustrate the paper. 

Originality / Novelty :
. The question set in this paper is original.  
. The question is very well defined.  

Significance :
. The scientific content of this paper is correct for me and deserves to be published.  
. The hypotheses are correctly identified as such.  
. The results are appropriately presented. Particularly, the figure 6, which is essential, is carefully drawn. 
. The presented results are significant.  
. The technical quality of this paper is correct for me.  
. The conclusion is correctly justified and supported by the results.  
. Some limits of the results obtained have been identified, presented and discussed. This is very interesting.  
. I took interest and pleasure to read this paper.  

Quality of presentation :
. The abstract is clear and presents correctly the subject addressed in this paper.  
. This paper contains the basic sections of a scientific paper.  
. The subheadings used for the redaction of this paper make it clear.  
. This paper is clear, easy to follow and to read, and logically written.  
. There is a lack of technical data, that could easily be fixed by the authors.  
. The conclusion is argumented and clear enough.  

Scientific soundness :
. The subject addressed in this paper is relevant.  
. The study has been correctly designed, and is technically sound.  
. In my opinion, the analyses of the results are convincing.  

Intererest to the readers :
. In my opinion, this paper is very interesting and deserves to attract a wide readership, beyond the limits of the journal's readership.  

Overall evaluation :
. I think there is an overall benefit to publish this work.  
. This work provides an advance towards the current knowledge, clearly highlighted in the abstract.  
. The authors have addressed an original question, with smart experiments. 
. The English language quality and style of this paper are appropriate and understandable.

As a conclusion, my suggestion to the editor is to accept this paper for publication in Electronics.

References :
--------------
. 11 research reference. This is rather few. 
. An effort could be done on the bibliography of this paper. 

Typos / Comments / Remarks :
------------------------------------
. You do not precise what is the ECSS. European Cooperation for Space Standardization.
. Line 250 (x2): SN less than four and greater or equal three (4 < SN ≥ 3) --> SN less than four and greater or equal three (4 > SN ≥ 3).

Author Response

(The authors gave the same response as above.)

Reviewer 3 Report

The paper is not uninteresting for the engineering point of view, however, the proposed approach for the radiation hardness assurance has been widely investigated in device works in the last 10 years, such as the following references:

L. Hash,  M.  R.  Shaneyfelt,  F.  W.  Sexton and    P.    S.    Winokur,    "Radiation    hardness assurance  categories  for  COTS  technologies," 1997  IEEE  Radiation  Effects  Data  Workshop NSREC   Snowmass   1997.   Workshop   Record Held  in  conjunction with  IEEE  Nuclear  and Space Radiation Effects Conference, Snowmass Village, CO, 1997, pp. 35-40.

M. J. Campola, “Taking Smallsats to the Next Level - Sensible Radiation Requirements and Qualification That Won't Break the Bank”, Small satellite conference, 2018

L. Ladbury and M. J. Campola, “Bayesian methods for bounding  single-event  related  risk in  low-cost  satellite  missions,” IEEE  Trans. Nucl. Sci., vol. 60, no. 6, pp. 4464–4469, 2013.

L. Ladbury  and  B.  Triggs,  “A  bayesian approach   for   total   ionizing   dose   hardness assurance,” in IEEE  Transactions  on  Nuclear Science, 2011, vol. 58, no. 6, pp. 3004–3010.

The paper is more a guideline than a real scientific paper.

For my point of view, the work should be released to the space community for a space agency workshop.

Author Response

(The authors gave the same response as above.)

Round 2

Reviewer 3 Report

the revised paper is almost the same, my review won't change.

The paper is not uninteresting for the engineering point of view, however, the proposed approach for the radiation hardness assurance has been widely investigated in device works in the last 10 years, such as the following references: G.  L.  Hash, M.
R.  Shaneyfelt,  F.  W.  Sexton and P.    S. Winokur, "Radiation
hardness assurance categories  for  COTS technologies," 1997
IEEE  Radiation Effects  Data  Workshop NSREC Snowmass   1997.
Workshop Record Held  in  conjunction with  IEEE Nuclear  and
Space Radiation Effects Conference, Snowmass Village, CO, 1997,
pp. 35-40.

[1]    M. J. Campola, “Taking Smallsats to the Next Level - Sensible Radiation Requirements and Qualification That Won't Break the Bank”, Small satellite conference, 2018

R. L. Ladbury and M. J. Campola, “Bayesian methods  for bounding single-event  related  risk in  low-cost  satellite missions,” IEEE Trans. Nucl. Sci., vol. 60, no. 6, pp. 4464–4469, 2013.

R.  L.  Ladbury  and  B.  Triggs,  “A  bayesian approach for total   ionizing   dose   hardness assurance,” in IEEE Transactions on  Nuclear Science, 2011, vol. 58, no. 6, pp. 3004–3010.